# Impact of carbon dioxide removal technologies on deep decarbonization of the electric power sector

John E. T. Bistline [1✉] & Geoffrey J. Blanford[1]

Carbon dioxide removal technologies, such as bioenergy with carbon capture and direct air capture, are valuable for stringent climate targets. Previous work has examined implications of carbon removal, primarily bioenergy-based technologies using integrated assessment models, but not investigated the effects of a portfolio of removal options on power systems in detail. Here, we explore impacts of carbon removal technologies on electric sector investments, costs, and emissions using a detailed capacity planning and dispatch model with hourly resolution. We show that adding carbon removal to a mix of low-carbon generation technologies lowers the costs of deep decarbonization. Changes to system costs and investments from including carbon removal are larger as policy ambition increases, reducing the dependence on technologies like advanced nuclear and long-duration storage. Bioenergy with carbon capture is selected for net-zero electric sector emissions targets, but direct air capture deployment increases as biomass supply costs rise.

[1] Electric Power Research Institute, Palo Alto, CA, USA. ✉email: jbistline@epri.com

The Paris Agreement aims to limit global warming to well below 2 °C and to pursue efforts to limit it to 1.5 °C. The literature describing scenarios that meet these targets relies heavily on the availability of negative emission technologies, also referred to as carbon dioxide removal (CDR) technologies, that capture $CO_2$ from the atmosphere[1–3]. There are questions about the speed, scale, cost, and acceptability of different CDR measures[4], but such strategies could be valuable to neutralize residual emissions and to draw down cumulative $CO_2$ from historical activity, especially for more stringent targets[5]. The value and role of CDR options such as bioenergy with carbon capture and sequestration (BECCS), direct air capture (DAC), and afforestation have been investigated primarily using integrated assessment models (IAMs)[6–9]. However, IAMs do not have the technological, temporal, or spatial resolution that more detailed energy systems models have[10,11]. Although CDR technologies can impact power sector planning, few detailed long-term energy system or power sector models include CDR options[10,12,13] and have not yet been used to compare how the availability and cost of CDR technologies can impact electric sector outcomes.

Power sector decarbonization is regarded as an important pillar of the deep decarbonization of energy systems and the economy as a whole through electrification and electricity-derived fuels[14–17]. Numerous studies look at power sector decarbonization with models of the energy system or power sector with technological, temporal, and spatial detail[18], but analyses with CDR options are limited and typically include only BECCS[12,19,20]. There are no studies in the extant literature that include a portfolio of CDR technologies and systematically investigate potential investment and operational impacts of CDR availability.

This paper addresses these gaps by investigating the role of CDR on power sector outcomes under deep decarbonization scenarios for the USA. We include technical and economic characteristics of the two main CDR options being pursued at demonstration scales, BECCS and DAC[21,22], alongside representations of other low-carbon power sector options in the Regional Economy, Greenhouse Gas, and Energy (REGEN) model, a state-of-the-art model of power sector investments and operations[23]. We compare pathways with and without CDR technologies to explore effects of CDR availability on decarbonization portfolios and costs. Given uncertainty about technologies and policy, we also conduct sensitivity analysis to explore how these factors can influence costs and emissions.

Results suggest that adding CDR to a mix of low-carbon generation technologies lowers the costs of achieving $CO_2$ reduction goals. CDR impacts are more significant as policy ambition increases, reducing the dependence on technologies like advanced nuclear and long-duration energy storage. BECCS is preferred to DAC for a net-zero electric sector $CO_2$ target (conditional on the availability of affordable and sustainably managed bioenergy), although DAC deployment increases as biomass supply costs rise in scenarios with higher CDR demand. Key considerations governing the role of CDR are their relative costs and value vis-à-vis other low-/zero-$CO_2$ options. CDR options create net negative emissions flows that offset expensive last tons of abatement in the electric sector, allowing zero-$CO_2$ emissions targets to become net zero. Moreover, CDR can provide cost-effective abatement to balance residual emissions from hard-to-decarbonize nonelectric sectors such as industry and heavy transport. Our results illustrate not only the potential importance of CDR technologies in reaching decarbonization and net-zero goals but also highlight uncertainties in technological cost and performance, supporting policies, and public acceptance.

## Results

**Modeling deep decarbonization in the electric sector.** Relative to earlier studies of CDR deployment, we use a power sector capacity planning and dispatch model with detailed technological, temporal, and spatial resolutions, which are critical in representing variable renewables, energy storage, and dispatchable low-carbon technologies[11,24,25]. The electric sector model, REGEN, is fully documented elsewhere[23], so only summaries of key features and assumptions are provided here, in the "Methods" section, and in Supplementary Note 1. Under a given set of assumptions about policies, technologies, and markets, REGEN optimizes decisions about new generation investments, energy storage and CDR capacities, hourly system dispatch, $CO_2$ transport and storage, transmission capacity, and trade. The variant used in this analysis is a single-year static equilibrium model with hourly operations and capacity investments, which captures the unique characteristics of variable renewable energy and joint distributions between time-series variables like regional load, potential wind generation, and potential solar generation.

This analysis focuses on scenarios for US electric sector deep decarbonization. Technological cost and performance estimates come from the literature, EPRI's Integrated Technology Generation Options report[26], and expert elicitations. Capital costs are summarized in Fig. 1 with additional detail in Supplementary Note 1 and are based on 2050 projections, though the analysis could be interpreted as an earlier year with accelerated technological change and policy commitments. Note that capital costs are only one factor in determining the optimal mix of technologies. For example, although BECCS is the most expensive generation option, the potential value of the negative emissions flow it creates can make it competitive with lower-cost technologies.

All scenarios are run under three CDR availability conditions: no CDR, DAC Only, and DAC + BECCS. Cost and performance assumptions for DAC and BECCS are detailed in Table 1. The economic and technical characterization of DAC is based on a high-temperature liquid solvent configuration owing to its lower costs of net $CO_2$ removal relative to other designs, accounting for the natural gas used for its heating requirement and capture of flue gas $CO_2$[27]. Low-temperature solid sorbent designs require additional cost reductions to be competitive but have the potential for higher learning rates from modularity and lower energy consumption due to lower regeneration temperatures[28–30]. The levelized cost of net $CO_2$ removal for BECCS and DAC are compared in Supplementary Figs. 18 and 19, respectively. Although gross levelized costs of net $CO_2$ removal are lower for DAC given the assumptions here, a key difference is that BECCS produces firm negative-$CO_2$ electricity generation as a coproduct.

We examine scenarios across the following dimensions in this analysis:

(1) $CO_2$ policy: Specifically, a cap on national $CO_2$ emissions relative to 2005 levels is used, spanning from 70 to 140% reductions. The 70% cap is the first level with a binding $CO_2$ constraint in the model under reference assumptions, and the 140% cap is selected to approximate CDR levels that offset difficult-to-decarbonize economy-wide $CO_2$ emissions categories, as described in the "Methods" section. Additional state and federal policies and incentives (e.g., tax credits, portfolio standards) are excluded to examine least-cost portfolios without side constraints.

(2) Choice set of low-/zero-/negative-$CO_2$ technologies: Reference (i.e., all technologies in Fig. 1) and renewables only.

(3) Wind/solar/storage costs (Fig. 1): Reference (i.e., best guess based on anticipated research and development); breakthrough (i.e., 5% probability outcome).

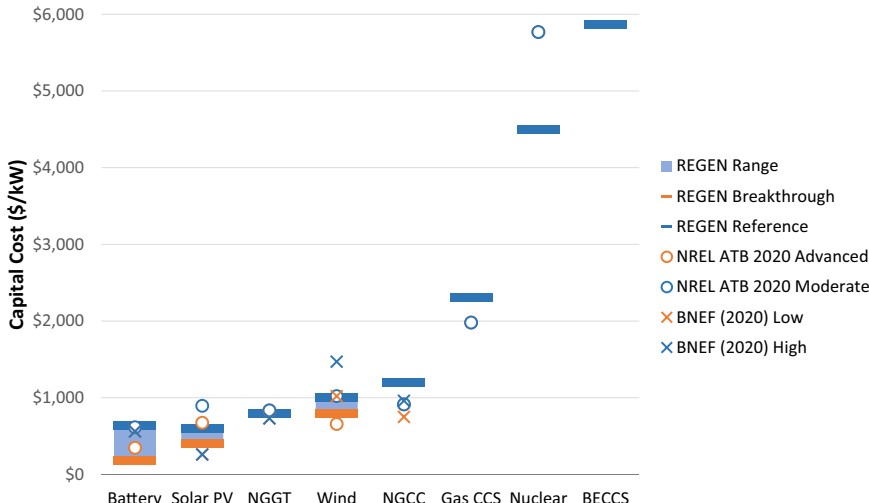

**Fig. 1 Capital cost assumptions (2018 USD per kW capacity) by technology in REGEN.** REGEN values come from EPRI's Integrated Technology Generation Options report[26] and expert elicitations. Reference assumptions are shown in blue, and breakthrough (i.e., low cost) assumptions are shown in orange for batteries, solar, and wind. Comparisons are provided with NREL's 2020 Electricity Annual Technology Baseline (ATB)[61] and BloombergNEF's "Levelized Cost of Electricity 2H 2020"[62]. Solar costs are expressed in $/kW$_{AC}$ terms. Battery costs in $/kW terms are shown for a 4-h duration system. Additional assumptions are discussed in the "Methods" section, Supplementary Note 1, and EPRI[23]. Source data are provided as a Source Data file.

**Table 1 Assumptions for CDR technologies in this analysis.**

|  | BECCS | DAC |
|---|---|---|
| Capital cost | $5870/kW ($568/t-CO$_2$/year net removal capacity) | $614/t-CO$_2$/year net removal capacity |
| Investment lifetime | 60 years | 30 years |
| Fuel cost | Based on FASOM-GHG model | Electricity prices are calculated endogenously in model; natural gas prices are scenario specific |
| CO$_2$ intensity | −1.18 t-CO$_2$/MWh | N/A |
| Biomass use | 14.8 MMBtu/MWh | N/A |
| Electricity use | N/A | 0.3 MWh/t-CO$_2$ |
| Natural gas use | N/A | 5.6 MMBtu/t-CO$_2$ |
| Fixed O&M cost | $177/kW/year ($17.1/t-CO$_2$/year) | $41/t-CO$_2$/year net removal capacity |
| Variable O&M cost | $19.4/MWh ($16.4/t-CO$_2$/year) excluding fuel and CO$_2$ transport/storage | $8/t-CO$_2$ excluding heat, electricity, and CO$_2$ transport/storage |
| Availability factor | Monthly values range from 60 to 80% | 90% |
| CO$_2$ transport/storage cost | Endogenous; varies by location | Endogenous; varies by location |
| Electricity coproduct | 0.85 MWh/t-CO$_2$ | N/A |

Reference BECCS assumptions are based on Johnson and Swisher[32], and DAC assumptions are based on Larsen et al.[31]. All values are expressed in 2018 US dollars. DAC parameters are normalized to net removal at the plant per metric ton of CO$_2$.

(4) DAC costs: Reference DAC capital costs ($614/t-CO$_2$/year) and a low-cost sensitivity ($107/t-CO$_2$/year) come from Larsen et al.[31], which is consistent with the low-cost scenario from Fasihi et al.[28]. These scenarios test the sensitivity of results to DAC costs given uncertainty about learning rates, policy support, and global deployment[28].

(5) BECCS cost and heat rate: Reference capital costs ($5870/kW) and higher-/lower-cost sensitivities ($10,000 and $3250/kW, respectively) are included. The reference heat rate (14.8 MMBtu/MWh) is accompanied by sensitivities with higher and lower values (17 and 6.8 MMBtu/MWh, respectively). Reference cost and heat rate assumptions are based on Johnson and Swisher[32] and sensitivities come from the literature survey in the recent Energy Modeling Forum 33 study on Bio-Energy and Land Use[33].

(6) Biomass resource availability: Regional biomass fuel costs are represented as piecewise linear supply curves (Supplementary Fig. 2) and are derived from the Forest and Agriculture Sector Optimization Model with Greenhouse Gases (FASOM-GHG), as described in the "Methods"

section. In addition to these reference supply curves, high and low biomass availability sensitivities are conducted by increasing and decreasing (respectively) each supply step by 50% for each region.

(7) CO$_2$ storage infrastructure and costs: In addition to the reference model configuration with endogenous CO$_2$ pipeline capacity for interregional transport and region-specific storage costs (see "Methods" section), we conduct sensitivities that restrict pipeline development and that equate CO$_2$ storage costs across regions. These bookend scenarios test how the spatial allocation of CDR technologies may change based on CO$_2$ storage potential.

**Investment and generation changes with CDR.** Model results suggest that CDR deployment increases with more stringent CO$_2$ policies, but CDR technologies are only deployed for achieving electric sector reductions of 90% or higher relative to 2005 levels (Fig. 2). For CO$_2$ reductions up to 100%, BECCS is preferred to DAC when both options are available at their reference costs (given

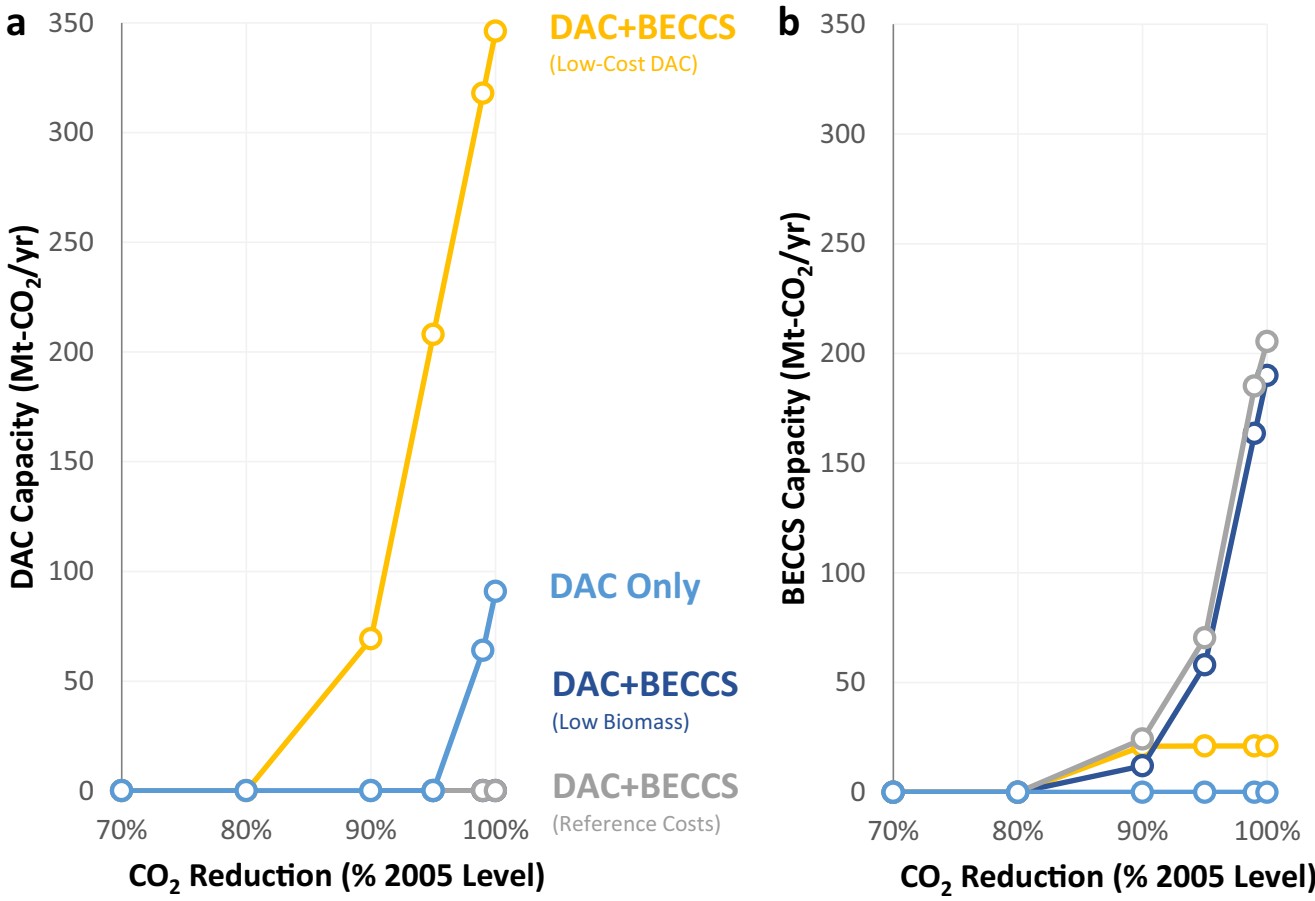

**Fig. 2 CO$_2$ removal capacity by electric sector CO$_2$ reduction level (% 2005 levels) and CDR availability scenario.** Panels show capacity for DAC (**a**) and BECCS (**b**). Scenario assumptions are described in the "Methods" section. Results are outputs from the REGEN model[23] detailed in the "Methods" section and Supplementary Note 1. Source data are provided as a Source Data file.

the assumptions described in the "Methods" section). However, a lower DAC capital cost increases deployment and decreases BECCS investment, as over 340 Mt-CO$_2$/year of DAC removal capacity (14.2% of 2005 electric sector CO$_2$ emissions levels) is deployed for the 100% CO$_2$ reduction scenario with a capital cost of $107/t-CO$_2$/year instead of the reference cost of $614/t-CO$_2$/year. When DAC is the only available CDR option, deployment is 91 Mt-CO$_2$/year for the 100% reduction scenario at reference costs.

CDR options create net negative emissions flows that offset expensive last tons of abatement in the electric sector and allow a positive emissions component to remain, enabling zero-CO$_2$ emissions targets to become net zero. Deployment of CDR increases the generation and capacity of natural-gas-fired units with and without carbon capture, in particular gas turbines which provide inexpensive capacity and operate at very low-capacity factors. Conversely, CDR availability decreases generation and capacity from advanced nuclear, renewables, and long-duration energy storage such as hydrogen to replace gas turbine capacity, especially under more stringent policy scenarios (Fig. 3). Relative changes in generation and capacity are shown in Supplementary Fig. 9. Without CDR, capacity of these technologies increases for CO$_2$ reductions greater than 80% (Supplementary Fig. 8). For instance, CDR lowers advanced nuclear capacity in the 100% cap scenario from 117 to 47 GW with DAC + BECCS (73 GW with DAC only). Overall, CDR availability makes least-cost capacity and generation portfolios less sensitive to the abatement target. Total BECCS capacity under the 100% cap is 0–41 GW for a range of assumptions about capital costs and heat rates (Supplementary Fig. 14).

Some analysts have suggested that current trends are enough to make 100% renewables systems least cost on their own. These scenarios illustrate that, even with optimistic cost assumptions, this equilibrium is not necessarily the case even though variable renewables are likely to comprise large shares of the power sector mix. Wind and solar comprise half of national generation without policy support (Supplementary Fig. 8). The marginal value of variable renewable profiles declines with penetration, so extensive wind and solar is accompanied by balancing resources. In addition to long-duration storage, seasonal balancing can also be provided by nuclear, dispatchable renewables, carbon-capture-equipped capacity, or even gas turbines with very low annual emissions. Least-cost decarbonization pathways differ by region and policy stringency, even as wind and solar shares generally rise with more stringent CO$_2$ limits in many regions (Supplementary Figs. 10 and 11).

CDR availability has significant impacts on the size and composition of energy storage deployment (Fig. 4). Energy storage is needed not only for integrating renewables but also to decarbonize the power sector by lowering natural gas consumption (as described in Supplementary Note 2). In particular, longer-duration storage (including electrolyzers, hydrogen storage, and hydrogen turbines) is limited when CDR is available, but otherwise entails a nonlinear increase with higher decarbonization. CDR does not play a role when the 100% cap is met through renewables only (RPS in Fig. 4). Note that battery storage deployment is high under all scenarios but depends on policy and technological cost assumptions more than CDR availability.

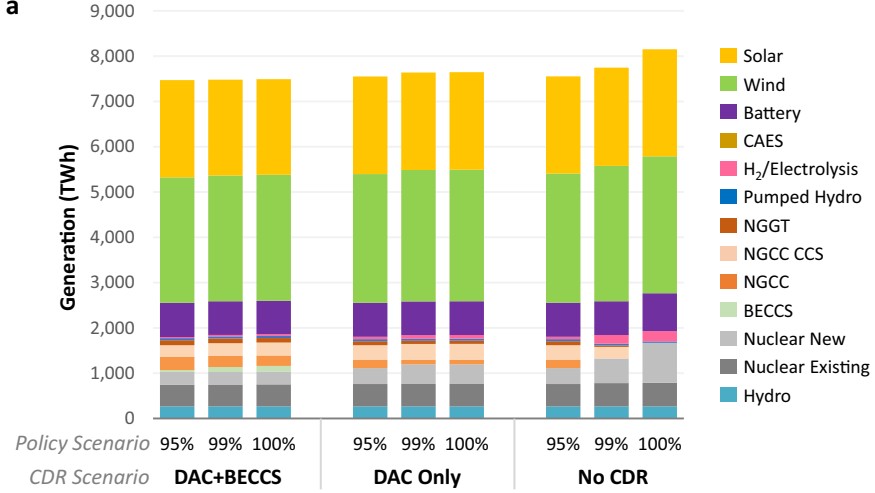

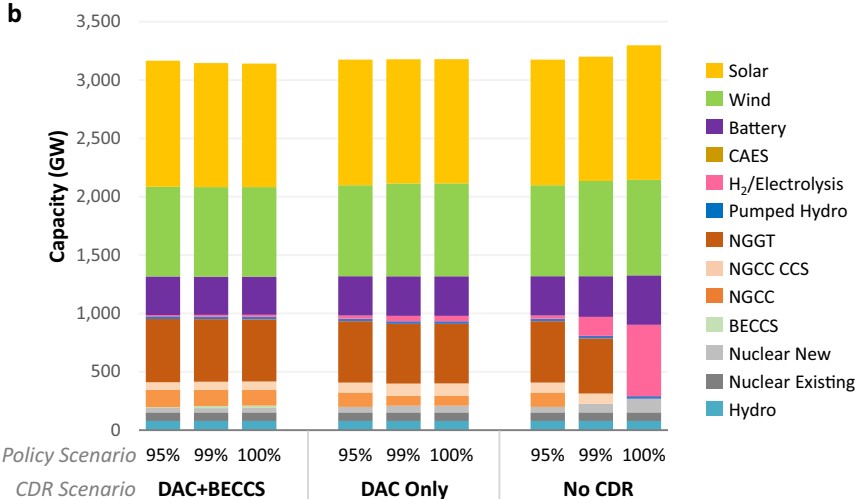

**Fig. 3 National electric sector results by technology and scenario.** Panels show the impacts of CDR on generation (**a**) and capacity (**b**). Cap refers to the electric sector $CO_2$ cap policy in terms of percentage reductions relative to 2005 levels. The top panel shows gross discharge from batteries and hydrogen. Scenario assumptions are described in the "Methods" section. Results are outputs from the REGEN model[23] detailed in the "Methods" section and Supplementary Note 1. Source data are provided as a Source Data file.

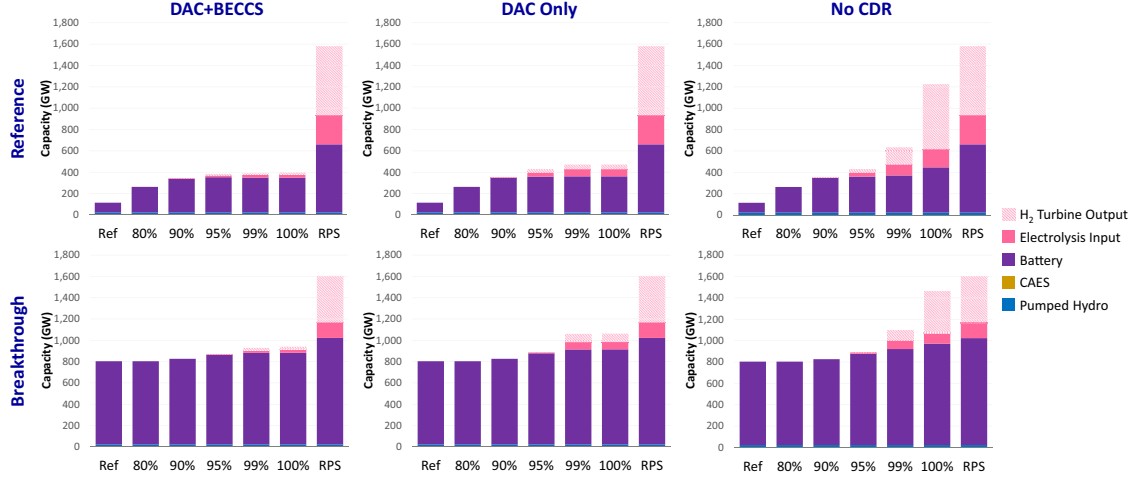

**Fig. 4 Energy storage capacity by technology across different electric sector $CO_2$ reduction targets (% reductions relative to 2005 levels).** Sensitivities include different renewable cost assumptions (rows) and CDR availability (columns). Ref refers to the reference scenario without $CO_2$ policy, and RPS refers to the zero-emissions scenario with renewables only. Source data are provided as a Source Data file.

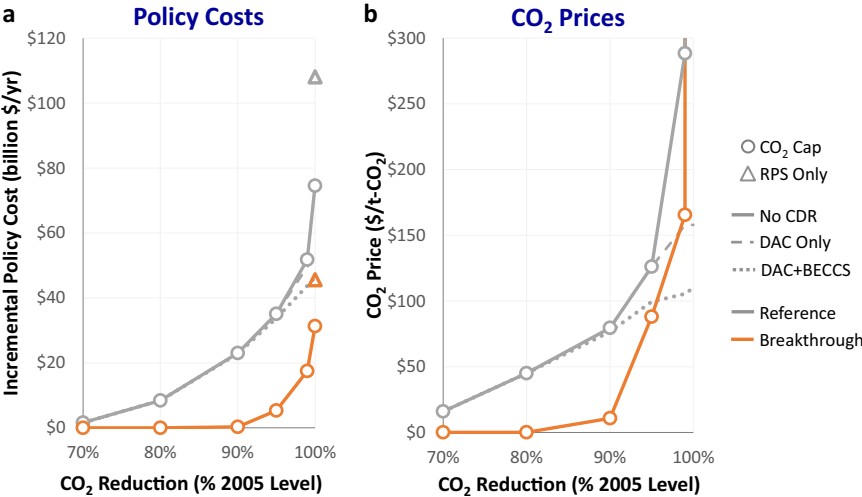

**Fig. 5 Economic impacts across different electric sector $CO_2$ reduction targets (% 2005 levels) and assumptions about CDR availability.** Panels show incremental policy costs (**a**) and $CO_2$ allowance prices (**b**). RPS only limits the choice set of eligible technologies to renewables and energy storage only. Incremental policy costs are the difference between annualized electric sector costs in the scenario with $CO_2$ caps and a reference scenario without policy. The vertical axis on (**b**) is truncated and omits 100% $CO_2$ reduction cases with reference and breakthrough technology assumptions, which have $CO_2$ prices of $849,000/t-$CO_2$ and $774,000/t-$CO_2$, respectively. Source data are provided as a Source Data file.

**Economic impacts of CDR**. CDR technologies can lower costs of achieving $CO_2$ targets by placing a ceiling on marginal abatement costs. Without CDR, decreasing returns lead to a convex marginal abatement cost curve (Fig. 5), as capital-intensive technologies with increasingly low utilization rates are deployed. The non-linear cost increases near 100% decarbonization without CDR hold even with significant cost reductions for renewables and battery storage, though the magnitude and slope depend on technological cost and availability. CDR lowers investments in capacity with high marginal abatement costs by providing emissions headroom for gas units that can provide these services at lower costs.

Total electric sector cost savings from CDR availability are greater as policy ambition increases (Fig. 5, left). Policy cost savings for the 100% cap with DAC only (DAC + BECCS) are $21.2 billion per year ($28.3 billion/year). Cost structures shift toward upfront capital expenditures for deeper decarbonization targets and away from fuel costs, and these higher investment costs drive incremental policy costs beyond reference expenditures without $CO_2$ policy, especially with very high renewables and energy storage shares (Supplementary Fig. 21). CDR availability has a larger impact on marginal cost savings, measured in terms of the $CO_2$ allowance price, than on total costs, and the flattening of the abatement cost curve with CDR leads to a linear increase of total costs in abatement effort (rather than a nonlinear increase without CDR). CDR availability lowers costs, but having both DAC and BECCS is only slightly lower cost than DAC alone. Note that, regardless of CDR availability, including dispatchable/firm low-carbon generation in the choice set lowers the cost of power sector decarbonization[34–36]. The cost premium of renewables only decarbonization is $33.5 billion per year (44.9%) higher than technology-neutral decarbonization without CDR ($14.3 billion per year or 45.7% higher under breakthrough assumptions). The renewables only generation and capacity mixes are shown in Supplementary Fig. 12.

A key dimension of cost savings from CDR availability is that DAC and BECCS replace low-capacity-factor assets with higher-utilization ones (Supplementary Fig. 15). DAC is only deployed under conditions with high enough $CO_2$ revenues to run with

very high capacity factors, and while it is possible to operate as a flexible load, the economic incentives for deployment do not align with such operational profiles, except for the highest demand hours. Similarly, although electricity sales help its economics, BECCS is valuable for its carbon sequestration, which leads to high capacity factors that are limited chiefly by assumed seasonal availability.

**Sensitivity on CDR demand for nonelectric residual emissions**. Additional sensitivities were conducted with net negative emissions targets for the power sector including DAC, which are consistent with modeled pathways for 1.5 °C with low overshoot[2]. Net negative emissions through CDR may be important for off-setting residual emissions for difficult-to-decarbonize end uses and sectors such as high-temperature industrial processes, aviation, shipping, and nonenergetic emissions[3,37]. Moreover, these sensitivities illustrate how higher CDR deployment may impact power sector planning. Emissions reductions range from 70 to 140% below 2005 levels (729 to −972 Mt-$CO_2$/year). The 140% cap is selected to approximate the CDR levels needed to offset difficult-to-decarbonize economy-wide $CO_2$ emissions categories, as described in the "Methods" section. The negative emissions from DAC and BECCS in this scenario lead to approximately net-zero economy-wide $CO_2$ emissions (Supplementary Fig. 7).

The level of the $CO_2$ reductions determines the portfolio of CDR technologies deployed (Fig. 6). BECCS is preferred to DAC through the net-zero (100%) $CO_2$ reduction target, but increasing biomass feedstock costs eventually make DAC more attractive at the margin for high-CDR-demand scenarios. Supplementary Fig. 17 illustrates DAC and BECCS deployment across different levels of policy stringency. With reference costs, BECCS deployment saturates at 110% $CO_2$ reductions (−243 Mt-$CO_2$/year) as marginal biomass feedstock costs increase, and DAC becomes the least-cost CDR technology for additional emissions reductions. This crossover point is reached at 105% (−121 Mt-$CO_2$/year) reductions with low biomass resource availability and 90% reductions (+243 Mt-$CO_2$/year) with low DAC costs.

1050 Mt-$CO_2$/year CDR is used in the 140% reduction scenario: 79.6 Mt-$CO_2$/year removal compensates for fossil

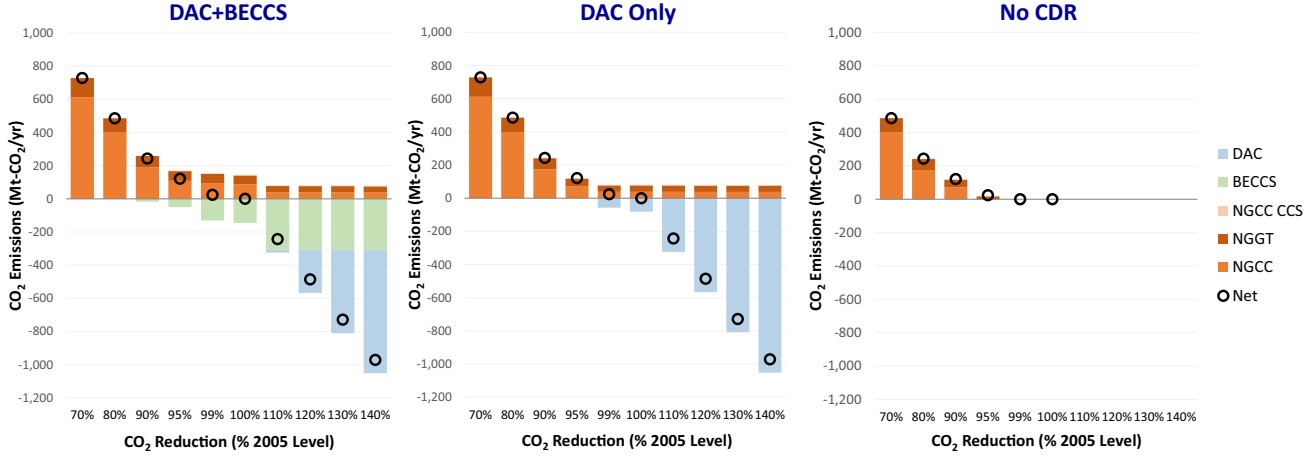

**Fig. 6 CO₂ emissions by technology across different electric sector reduction targets (% reductions relative to 2005 levels) and assumptions about CDR availability.** Net emissions from bioenergy with carbon capture and sequestration (BECCS) and direct air capture (DAC) are negative, while emissions from natural-gas-fired combined cycle (NGCC), CCS-equipped gas (NGCC CCS), and gas turbines (NGGT) are positive. Note that emissions reductions beyond 100% are not feasible in the cases without CDR. Source data are provided as a Source Data file.

generation in system, while the remainder offsets emissions in other sectors. CDR demand for scenarios with greater than 100% CO₂ reductions leads to higher electricity demand but modest shifts in generation shares (Supplementary Fig. 16).

Costs of emissions reductions beyond 100% increase linearly in abatement (Supplementary Fig. 22) owing to CDR technologies flattening the marginal abatement cost curve by providing a backstop mitigation option (Supplementary Fig. 23). CDR helps to achieve greater emissions reductions with equivalent expenditures or to reach the same emissions with lower costs. For instance, for the same expenditure as an all-renewable system, an electric sector with CDR deployment can achieve almost 20% more CO₂ reductions relative to 2005 levels (Supplementary Fig. 22).

The impact of DAC on electric load is small even under high DAC deployment scenarios relative to other factors such as transport electrification, industrial electrification, and net losses from energy storage (Fig. 7). DAC consumes 24.8 TWh/year (322 TWh/year) in the 100% (140%) CO₂ cap case with DAC only (with net CO₂ removals of 81.1 Mt-CO₂/year and 1050 Mt-CO₂/year, respectively, as shown in Fig. 6), which is 0.42% (5.39%) of projected end-use electricity demand. In fact, net energy storage losses in the 100% CO₂ cap case without CDR (548 TWh/year) are over an order of magnitude higher than DAC electricity use in the 100% DAC Only case (24.8 TWh/year), since gas turbines are replaced with hydrogen and electrolysis with low roundtrip efficiencies (Fig. 3). Although there are economic and regulatory issues to address regarding CDR deployment, this analysis suggests that power sector integration issues are manageable for the stringencies examined here. For instance, biomass consumption from BECCS deployment under the 100% CO₂ cap is ~1.81 quads (Supplementary Fig. 20). 2019 biomass production in the USA totaled 4.82 quads[38], so BECCS at this scale represents a nontrivial but likely manageable increase.

**The economic geography of CDR: geospatial sensitivity analysis to cost and market uncertainties.** The spatial distribution of CDR technologies depends on factors with considerable regional variation such as biomass availability, suitable geologic CO₂ storage sites, and technological cost and availability. The sensitivities in this section vary these factors to explore effects on the spatial allocation of CDR.

The geographical distribution of CDR deployment across sensitivities is shown in Fig. 8. BECCS deployment is spread across a greater variety of regions relative to DAC, and the highest BECCS potential occurs in regions such as the Gulf, Southeast, Ohio Valley, and portions of the Midwest. This spatial deployment generally aligns with BECCS economic and technical potential estimates[39,40], though our modeling includes not only biomass availability and CO₂ storage but also endogenous CO₂ pipeline capacity and electric sector capacity planning and dispatch. BECCS plants are primarily located in regions with higher biomass availability, especially since biomass transport costs are high due to its low energy density. Biomass costs account for large fractions of the levelized cost of BECCS (Supplementary Fig. 18) and, along with CO₂ storage costs, represent factors with higher degrees of regional variation. Low and high biomass resource availability alter the total deployment of BECCS, but the spatial distribution of BECCS capacity is similar across model regions. Constraining CO₂ pipelines only impacts New England, since it is the only model region without geologic storage capacity (Supplementary Fig. 5). The fraction of available biomass resource utilized across scenarios and regions is shown in Supplementary Fig. 20. Biomass consumption for BECCS in most regions exhausts the first few supply steps before the piecewise linear supply curves increase from $5/MMBtu to $9/MMBtu (Supplementary Fig. 2).

DAC deployment is less evenly distributed across regions (Fig. 8). Spatial variability is highest for CO₂ storage costs and electricity prices (Supplementary Fig. 20), and the regions with highest DAC capacity (the South Atlantic, California, MISO South, and Texas) are ones with lower combined costs across these dimensions. The spatial allocation of CDR deployment is determined not only by regional variation in costs but also value; however, the value of carbon removal for DAC is the same across regions due to the scenario assumption of a national CO₂ cap.

Note that there is potential spatial heterogeneity not accounted for in this analysis that could influence siting decisions for CDR technologies, including public acceptance, potential for CO₂ utilization, state-level policies and incentives outside the power sector (e.g., low-carbon fuel standard eligibility), heat costs, and CO₂ capture and storage outside of the power sector.

**Operational flexibility of CDR technologies.** Earlier results assume that BECCS capacity is dispatchable, meaning that hourly

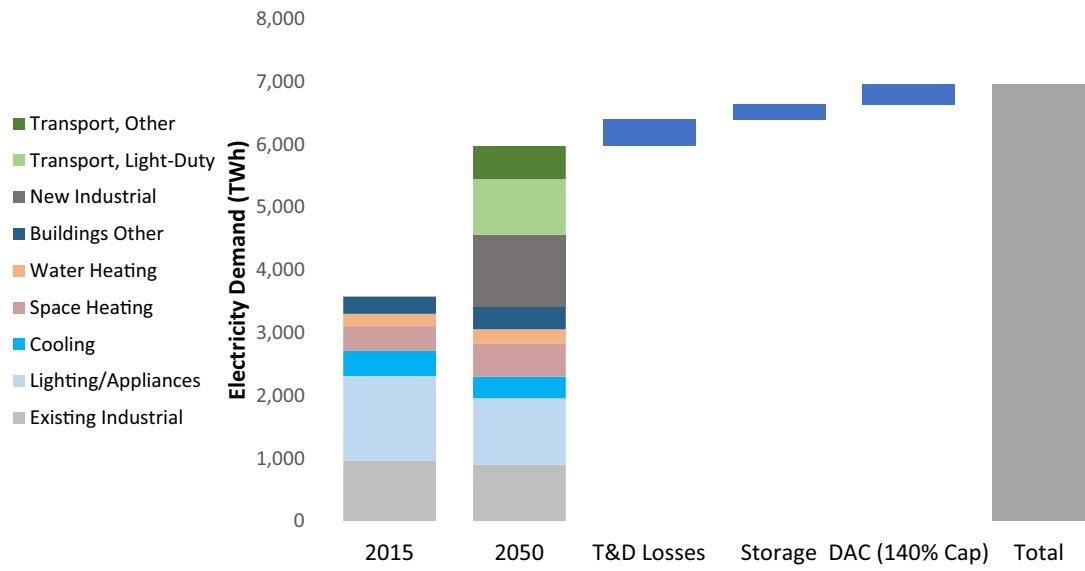

**Fig. 7 Electricity demand decomposition by end-use application and supply losses.** 2015 load is compared with simulated 2050 demand in the 140% electric sector CO$_2$ cap scenario with DAC only. Storage losses include net losses from all energy storage technologies (batteries, pumped hydro, and hydrogen via electrolysis). Electricity demand comes from the REGEN end-use model with assumptions described in the "Methods" section. Source data are provided as a Source Data file.

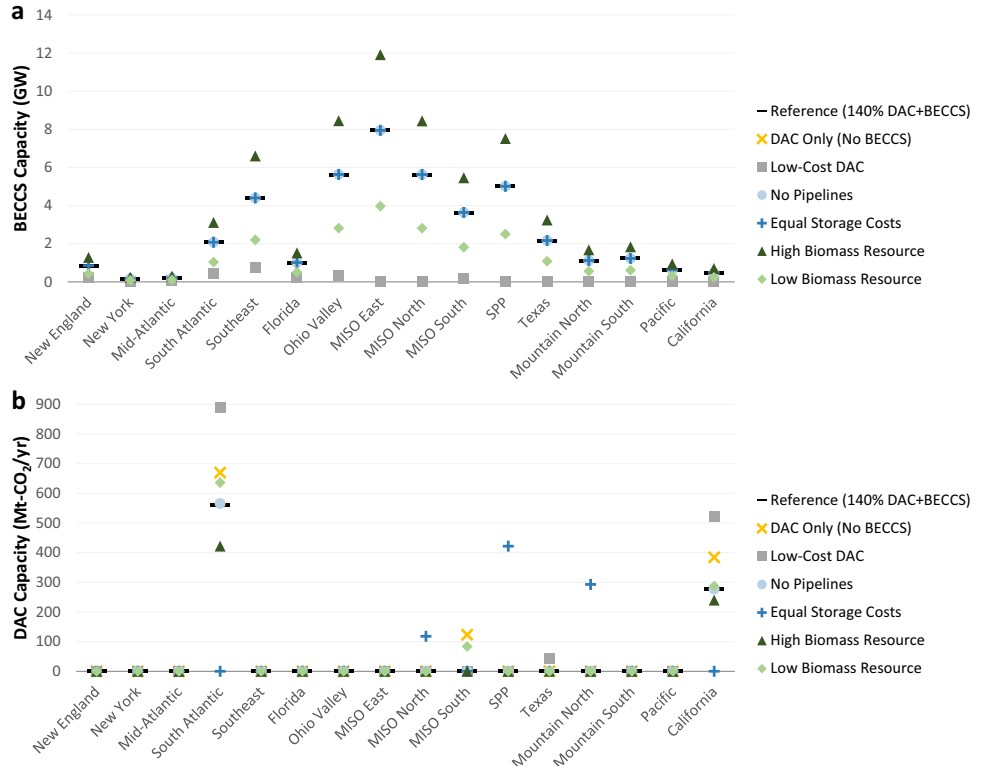

**Fig. 8 Regional deployment of CDR technologies across scenarios.** Panels show BECCS (**a**) and DAC (**b**) capacity by CDR availability scenario under 140% electric sector CO$_2$ reductions (relative to 2005 levels). Regional definitions are shown in Supplementary Fig. 1. Source data are provided as a Source Data file.

output can be adjusted based on market conditions and unit availability. Plants will likely be designed with flexibility in mind[41], but it is unclear how less flexible plant designs could alter the economics of BECCS, as the technology operates with very high capacity factors even when plants are assumed to be dispatchable (Supplementary Fig. 15). We test these implications through a sensitivity that considers BECCS to be a must-run

resource under the 140% CO$_2$ reduction policy. BECCS capacity remains at 42.2 GW nationally in both the dispatchable and must-run/inflexible cases. Changes are limited across this operational flexibility sensitivity, because constraints that limit BECCS dispatch are not binding in nearly all instances: The value of carbon removal and electricity production exceeds the short-run dispatch costs of BECCS plants. In part, the high utilization

of BECCS plants is due to their carbon removal value being higher than the value of electricity (as illustrated in Supplementary Figs. 18 and 19), which is mirrored in other studies[19]. Under typical assumptions about heat rates and emissions factors, a BECCS unit would receive approximately a \$1/MWh payment for carbon removal for each \$1/t-$CO_2$ carbon price.

DAC capacity is built under conditions where the value of carbon removal is high enough to run with very high capacity factors. Although some have speculated that DAC can help variable renewable integration by absorbing excess supplies during high-output periods[42], such operational profiles do not align with the economic incentives for DAC investment. The scenarios in this paper suggest that DAC utilization rates tend to be close to 8000 h per year (Supplementary Fig. 15) instead of hundreds of hours per year if DAC were powered only during periods of curtailed renewables (i.e., very low or zero power prices). The high-output operations of DAC imply more constant heat and electricity supplies, which is reinforced by Supplementary Figs. 9 and 16. Note that, since DAC is powered by the grid and not a dedicated electricity supply, it is difficult to isolate the precise marginal generation mix powering DAC; however, it is only deployed in the context of a deeply decarbonized generation mix.

Overall, the results illustrate how BECCS and DAC (despite their potential flexibility) tend toward high-utilization operations and can be compatible with a range of low-carbon and high-renewable systems. BECCS and DAC are small enough components of regional power systems in these scenarios that their investment and operational dynamics are influenced less by market fluctuations from variable renewables relative to other resources.

## Discussion

This analysis demonstrates that CDR availability can materially impact power sector decarbonization pathways and help to lower costs associated with $CO_2$ emissions reductions, especially for higher mitigation levels. CDR provides flexibility in meeting net-zero targets, reducing the dependence on more costly abatement options and avoiding the overdependence on any single emerging technology. Without CDR, electric sector abatement costs increase sharply beyond 90% $CO_2$ reductions, even with significant cost reductions for renewables and energy storage. Including CDR technologies in the choice set lowers the cost of electric sector decarbonization, complementing extensive conventional mitigation as part of cost-minimizing pathways to reach net-zero targets.

We demonstrate how the cost savings of CDR is insensitive to various robustness checks. The scenarios investigated here demonstrate that CDR technologies could be part of a least-cost decarbonization strategy under a range of plausible deep decarbonization scenarios, though the mix of CDR technologies deployed is sensitive to cost assumptions and biomass resource availability. This finding should encourage modeling teams and resource planners to incorporate BECCS, DAC, and other CDR options into the technology choice set in their modeling, especially as deep decarbonization and net-zero targets are pursued by countries, subnational jurisdictions, and companies. Large uncertainties associated with CDR technologies should persuade analysts to conduct a wide range of sensitivities to understand how cost, performance, and other parameters can influence decisions.

Across the range of sensitivities examined, nascent technologies are used in some degree as part of electric sector least-cost decarbonization portfolios, but the availability of CDR can provide optionality if some of these technologies exhibit unforeseen

technological hurdles or public acceptance issues by limiting their rate of deployment[43]. Moreover, CDR has a greater likelihood of additionality vis-à-vis traditional offsets as countries, subnational jurisdictions, and companies pursue net-zero goals[44]. Despite uncertainty about technologies to reduce the last 10–20% of $CO_2$, a robust finding from these sensitivities is that renewables comprise the backbone of electric sector decarbonization scenarios, even if they are not 100% of the generation mix. Wind and solar are half of the national generation share in the reference cost scenario without policy.

The analysis also highlights how CDR availability lowers mitigation costs by substituting lower utilization resources such as long-duration energy storage with higher capacity factor CDR options. Although the existing literature has mentioned the role of long-duration storage in very high-renewable and low emissions systems[45], we are the first analysis to illustrate how CDR options can enable gas turbines as cost-effective substitutes to long-duration energy storage technologies for low-capacity-factor firm capacity.

We show how CDR technologies can have economic and environmental value in the context of sectoral and national targets as well as global targets, even though most literature to date has focused on the latter. We also demonstrate how the value proposition of DAC suggests that it is unlikely to be the flexibility resource some have speculated to help renewable integration. Electricity consumption from DAC—the quantification of which has been emphasized by other studies—is shown to be small relative to expected electrification and losses from energy storage even under high DAC deployment scenarios (less than 5% of total load for most scenarios examined here).

Given the limitations of the modeling scope, this analysis should be supplemented by qualitative and quantitative analyses of CDR deployment in economy-wide decarbonization scenarios. Many CDR issues are outside of the scope of this analysis such as RD&D strategy and financing first-of-a-kind units[46], policy design[47], lifecycle emissions associated with biomass production[4], and geological characterization of $CO_2$ storage and site selection[48]. In addition to the economic dimensions examined in this analysis, the desirability of DAC pathways relative to BECCS (and other low-/zero-/negative-$CO_2$ technologies) may be influenced by differences in land use change[4,49], water demand[10], lifecycle environmental impacts[10,50], nonelectric decarbonization interactions[3], and innovation spillovers and higher learning rates for modular technologies[29], all of which increase the favorability of DAC. Quantifying these tradeoffs is left for future work.

## Methods

**Optimization model.** The analysis is based on scenarios conducted in EPRI's US REGEN model, which features an electric sector capacity planning and dispatch model linked to an end-use model with technological, temporal, and spatial detail[23]. REGEN is fully documented in EPRI[23], so only summaries of key features and assumptions are provided here.

The REGEN electric sector model is formulated as a linear program that minimizes the net present value of total system costs subject to technical and economic constraints under given scenario assumptions. REGEN includes endogenous capacity planning and dispatch with joint investment decisions in generation, energy storage, transmission, and CDR capacity. The variant used in this analysis is a single-year (2050) static equilibrium model with capacity investment and hourly dispatch. In this mode, REGEN adds new capacity for most of the system (i.e., greenfield investment), inheriting only endowments of existing hydropower, nuclear, and interregional transmission. The use of the static model allows the analysis to represent hourly operations and capacity investments faithfully, something that is not currently possible in long-horizon dynamic models without chronology, which use a few representative hours for a single year[51]. The literature has demonstrated that a model must be able to capture the declining economic value of variable renewable energy at higher penetration levels and ability of system resources like energy storage to mitigate these effects, which are captured in REGEN[23,52]. REGEN represents a broad range of technologies. Technological cost and performance estimates come from the literature, EPRI's Integrated Technology

Generation Options report[26], and expert elicitations. Capital costs are summarized in Fig. 1 and are based on 2050 projections. In addition to new investments, the model includes existing capacity endowments of pumped hydropower, conventional hydropower, nuclear (units that would be online in 2050 with 80-year license extensions, which is ~73 GW), and interregional transmission capacity. Data to characterize the existing fleet come from ABB Energy Velocity. REGEN aggregates US states in the 16 regions shown in Supplementary Fig. 1.

REGEN represents a range of energy storage technologies such as batteries, compressed air energy storage, existing pumped hydro, and hydrogen via electrolysis. For batteries, charging and discharging capacities of the inverter are assumed to be equal, and the model endogenously selects battery storage investment and system configurations (i.e., ratio of energy capacity to power capacity) based on cost structure assumptions from EPRI[53], which are shown in Supplementary Fig. 3. Battery costs in Fig. 1 are based on a 4-h lithium-ion system. REGEN includes energy storage market participation for energy arbitrage, capacity value, ancillary services (namely, operating reserves when specified), and interregional transmission deferral. Existing pumped hydro storage is assumed to have energy storage capacity of 20 h at nominal power and is limited to current installed capacity. For hydrogen storage pathways, the model independently optimizes the capacity of hydrogen production via electrolysis, hydrogen storage, and generation from hydrogen turbines. The assumed electrolysis capital costs of $200/kW are at the lower range of current estimates (Supplementary Fig. 4); the cost of electricity input is endogenously determined from the grid mix. Costs of hydrogen storage are assumed to be $50/MMBtu, which are similar to storage cost estimates for salt caverns[54].

Hourly regional renewable output and resource potentials are based on analysis and data by EPRI, AWS Truepower, and NASA's MERRA-2 dataset and provide synchronous time-series values with load. Variability is modeled using gridded hourly data from NASA's MERRA-2 dataset, which provides key meteorological variables such as wind speed, solar irradiation, and temperature. For wind technologies, wind speed at hub height is translated into power output based on assumed power curves for a range of turbine technologies. Wind output profiles for given regions and resource classes vary by vintage based on an assumed mix of turbine type and hub height, as detailed in EPRI[23]. For solar technologies, REGEN considers three types of central station solar PV technologies (fixed-tilt crystalline silicon, single-axis tracking, and double-axis tracking), concentrated solar with endogenous determination of thermal storage, and fixed-tilt rooftop solar PV. Solar output profiles are derived from gridded hourly radiation flux data from MERRA-2. Diffuse and direct irradiance are translated into output for a variety of solar photovoltaic technologies that are specified in terms of the orientation and tilt of the panels. Captured energy at the panel is adjusted for temperature impacts on module efficiency, nonlinear inverter losses, and a gross de-rating factor reflecting a range of factors not otherwise captured. Additional detail is provided in Section 2.4.1 of the full REGEN documentation[23]. Hourly profiles used in the model solution are based on a single representative year (2015 for these experiments), and the same underlying meteorology and temperatures are used in the end-use model to develop hourly load shapes (e.g., for electric space heating in residential and commercial buildings) to avoid dampening variance through multiyear averaging. While consideration of multiple weather years may reveal more extreme events, there are significant wind droughts observed in the sample year, reinforcing the importance of energy storage and firm resources for balancing. Moreover, the model includes a reserve requirement that firm capacity (excluding variable renewables) exceed peak load in each region, suggesting the results are relatively robust to extended wind and solar droughts. Additional detail on wind and solar resource assumptions and technology characteristics is provided in Section 2.4 of the detailed REGEN documentation[23].

Hourly load profiles come from the REGEN end-use model, which characterizes the economic and behavioral incentives for technology adoption and captures heterogeneity across households, industries, and regions[14]. To reflect the deep decarbonization context of the power sector sensitivities, the end-use model scenario assumes federal $CO_2$ pricing of $50/t-$CO_2$ in all sectors and regions beginning in 2020, escalating at the model's discount rate of seven percent per year. The national average power producer price of natural gas is assumed to be $4/MMBtu (in 2018$).

Cross-regional exchange of electricity in a given hour is constrained by net transfer capacities of transmission between regions, which can change over time as new investments are made. Base year interregional transmission capacity comes from the National Renewable Energy Laboratory's ReEDS model. Transmission between regions can be endogenously added with an assumed cost of $3.85 million per mile for a notional high-voltage line to transfer 6400 MW of capacity. The model also includes a $4/MWh transaction cost for power transmitted between any two regions, which functions as an artificial proxy for operational barriers to interregional balancing not captured by the model's spatial resolution. Interconnection costs for utility-scale wind (solar PV) are $250/kW ($100/kW) across all regions. Emissions factors do not include lifecycle-related emissions with generation technologies or fuels.

Incremental policy costs (as shown in Fig. 5) are the difference between electric sector costs in the scenario with $CO_2$ caps and a reference scenario without $CO_2$ policy. Electric sector costs include investment, fuel, operations, and maintenance costs for generation, bulk transmission, energy storage, and carbon removal assets, but exclude intraregion transmission and distribution costs.

There are a few caveats to keep in mind when interpreting the analysis. First, the focus on the deep decarbonization end point using a static optimization framework abstracts from the transition path, and considering an intertemporal optimization could impact the cost and value of different end points. Second, the analysis does not explicitly model operational constraints, ancillary services markets, or sub-hourly/sub-state detail[55]. Finally, demand is fixed in this analysis across scenarios (see Supplementary Note 1) but could be an additional source of system flexibility[56,57].

**Modeling CDR technologies.** A CDR portfolio could include BECCS, DAC, afforestation/reforestation, ocean fertilization, enhanced weathering of minerals, and biochar[47]. Since BECCS and DAC are the only CDR options currently being pursued at demonstration scale[21,22], we focus on these technologies in the analysis. Based on Johnson and Swisher[32], BECCS has a capital cost of $5870/kW and heat rate of 14.8 MMBtu/MWh (see Table 1). The capital cost of BECCS translates into $568/t-$CO_2$/year net removal capacity given the net removal of 10.3 t-$CO_2$/kW/year. Sensitivities for capital costs and heat rates come from the literature survey in the recent Energy Modeling Forum 33 study on Bio-Energy and Land Use[33].

The economic and technical characterization of DAC (Table 1) is based on a high-temperature liquid solvent configuration owing to its lower costs of net $CO_2$ removal relative to other designs, accounting for the natural gas used for its heating requirements and capture of flue gas $CO_2$[27]. Low-temperature solid sorbent designs require greater cost reductions to be competitive but have a high potential for capital and maintenance cost reductions from modularity, economies of scale and learning-by-doing from mass production, and technical advances, which also can lead to lower energy requirements[28–30]. Based on Larsen et al.[31], reference DAC capital costs are $614/t-$CO_2$/year capture capacity, and a low-cost sensitivity is conducted at $107/t-$CO_2$/year (with commensurate percentage reductions in fixed and variable operations and maintenance costs). Most DAC articles focus on technical parameters and only a few have economic estimates, but the low-cost scenario used here from Larsen et al.[31] also aligns with the low-case from Fasihi et al.[28] in 2050. This analysis assumes 5.6 MMBtu/t-$CO_2$ heat requirement for DAC based on Larsen et al.[31], which is in the middle of the ranges given in Realmonte et al.[6] and Fasihi et al.[28]. The 30-year DAC lifetime is based on Larsen et al.[31] and Fasihi et al.[28].

REGEN models the transport of captured $CO_2$ to injection sites where it is stored in saline aquifers (Supplementary Fig. 5). $CO_2$ transport and storage costs vary by location and volume stored. Regional $CO_2$ storage capacity is limited based on estimates from the National Carbon Sequestration Database (NATCARB), which is populated from the U.S. Department of Energy's "*Carbon Storage Atlas: Fifth Edition*"[58]. Regions must invest in $CO_2$ injection capacity to enable storage, and once regional storage limits are reached, investments in interregional $CO_2$ pipeline capacity can be made. Regional storage constraints are not binding here except for regions without any suitable geological storage capacity (e.g., New England). The capital costs of CCS-equipped technologies include the cost of a 20-mile $CO_2$ pipeline that enables access to a dedicated injection site or large pipeline for interregional transport. Additional information about the $CO_2$ transport/storage formulation and assumptions in REGEN can be found in the detailed model documentation[23]. Note that captured $CO_2$ in this analysis is assumed to be stored; future work should examine how utilization of captured $CO_2$ could lower costs[59].

Regional biomass fuel costs are represented as piecewise linear supply curves (Supplementary Fig. 2). These supply curves are derived from the FASOM-GHG. FASOM-GHG is used to estimate regional supply curves over time for the delivered costs biomass resources for energy production in electric and nonelectric applications. FASOM-GHG endogenously accounts for food, feed, coproduct, and other bioenergy market feedbacks. The regional agricultural and forestry cellulosic biomass supply curves are inputs to the REGEN electric sector model. Curves are estimated at the state level and then aggregated in the model regions shown in Supplementary Fig. 1. Additional detail on the forestry and agricultural biomass supply modeling is provided in Appendix B of the detailed REGEN documentation[23].

**Scenario development and study design.** All scenarios are run under three CDR availability conditions to quantify impacts on power sector outcomes: no CDR, DAC only, and DAC + BECCS. We examine scenarios across the following dimensions in this analysis:

(1) $CO_2$ policy: Specifically, a cap on national $CO_2$ emissions (relative to 2005 levels) is used, spanning from 70 to 140% reductions. $CO_2$ caps on electric sector emissions are relative to 2005 levels of 2430 million metric tons of $CO_2$. The 70% cap is the first level with a binding $CO_2$ constraint in the model, and the 140% cap is selected to approximate the CDR levels needed to offset difficult-to-decarbonize economy-wide $CO_2$ emissions sources (e.g., iron/steel, cement, aviation, shipping). Davis et al.[37] indicate that ~15% of economy-wide $CO_2$ emissions fall into this category, and 15% of 2005 U.S. $CO_2$ emissions is roughly 40% of electric sector emissions. The specific $CO_2$ reduction levels considered in the analysis are 70, 80, 90, 95, 99, 100, 105, 110, 115, 120, 125, 130, 135, and 140%. The analysis conducts additional sensitivities to have higher resolution as the electric sector approaches 100%

emissions reductions (e.g., to characterize cost and investment changes). Market-based emissions policies such as the $CO_2$ emissions caps modeled here lower aggregate costs of reaching environmental targets by inducing additional abatement effort from polluters with lower marginal abatement costs[60]. Additional state and federal policies and incentives (e.g., tax credits, portfolio standards, technology carve-outs) are excluded from the analysis to examine least-cost portfolios without side constraints.

(2) Choice set of low-/zero-/negative-$CO_2$ technologies: Reference (i.e., all technologies in Fig. 1) and renewables only.

(3) Wind/solar/storage costs (Fig. 1): Reference (i.e., best guess based on anticipated research and development); breakthrough (i.e., 5% probability outcome).

(4) DAC costs: Reference DAC capital costs ($614/t-$CO_2$/year) and a low-cost sensitivity ($107/t-$CO_2$/year) both from Larsen et al.[31].

(5) BECCS cost and heat rate: Reference capital costs ($5870/kW or $568/t-$CO_2$/year net removal capacity) and higher-/lower-cost sensitivities ($10,000 and $3250/kW, which translate to $967/t-$CO_2$/year and $314/t-$CO_2$/year, respectively) are included. The reference heat rate (14.8 MMBtu/MWh) is accompanied by sensitivities with higher and lower values (17 and 6.8 MMBtu/MWh, respectively). Reference cost and heat rate assumptions are based on Johnson and Swisher[32]. Sensitivities come from the literature survey in the recent Energy Modeling Forum 33 study on Bio-Energy and Land Use[33] and use the highest and lowest reported conversion efficiency and capital cost from the literature.

(6) Biomass resource availability: In addition to the reference biomass supply curves, high and low biomass availability sensitivities are conducted by increasing and decreasing (respectively) each supply step by 50% for each region. These stylized cases explore the impact of alternate biomass availability assumptions on the regional deployment of BECCS and other clean technologies.

(7) $CO_2$ storage infrastructure and costs: In addition to the reference model configuration with endogenous $CO_2$ pipeline capacity for interregional transport and region-specific storage costs, we conduct sensitivities that restrict pipeline development and that equate $CO_2$ storage costs across regions. These bookend scenarios test how the spatial allocation of CDR technologies may change based on $CO_2$ storage potential.

## Data availability
Source data underlying all figures are provided as a Source Data file. All other nonproprietary data supporting this study are available from the corresponding author upon reasonable request. Source Data are provided with this paper.

## Code availability
The optimization code that supports the analysis within this paper is available from the corresponding author upon reasonable request.

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

## Acknowledgements

The authors would like to thank Nils Johnson, David McCollum, Steven Rose, David Young, and anonymous reviewers for their helpful comments. The authors wish to recognize the many subject matter experts who provided input to the analysis and assumptions, including John Larsen, Whitney Herndon, Erin Minear, Marcus Alexander, Robin Bedilion, Neil Kern, and Joseph Swisher. The views expressed in this paper are those of the individual authors and do not necessarily reflect those of EPRI or its members.

## Author contributions

Both authors developed the model, conceived the study, designed scenarios, and contributed to writing the article. J.E.T.B. led the modeling, data analysis, and article revisions.

## Competing interests

The authors declare no competing interests.
