## [Peer Review File · Nature Communications]

REVIEWER COMMENTS

Reviewer #1 (Remarks to the Author):

This is a well-written, rigorous paper on an under-studied topic of interest to broad readership: the impacts of carbon removal technologies (BECCS and DAC) deployment on deep decarbonization of the U.S. electric power sector. The model, REGEN, is well-developed and well-documented. The range of policy scenarios tested are reasonable. The results are of value to numerous decision-makers.

Nevertheless, I believe the representation of inputs, results, and analysis could be improved in many ways. Namely, there needs to be increased discussion of the geospatial aspects of deployment. Specifically, authors must consider and present analysis of variation of BECCS and DAC deployment and discuss factors with largest potential regional variation: geologic storage potential, regional biomass availability, and availability of low-cost renewable resources. These have been identified as potential drivers of variation in CDR deployment in prior studies (discussed below), but are noticeably lacking from the report, save for data reported in a single figure in the Appendix. I require this issue to be addressed prior to publication in Nature Communications.

I also have several other suggested revisions that I believe are within scope of a capacity expansion model with high spatial and temporal resolution such as REGEN. These revisions may produce additional insights of value to potential readers.

Required revisions:

-Geospatial representation of deployment

Authors must consider and present analysis of variation of BECCS and DAC deployment and discuss factors with largest potential regional variation: geologic storage potential, regional biomass availability, and availability of low-cost renewable resources.

Based upon the information provided in Figure Figure 22, "Regional deployment of BECCS and DAC capacity by CDR availability scenario under the 140% CO₂ reduction scenario." I believe this is a reasonable request.

Possible questions to be addressed by the analysis & discussion could include: are there regions of REGEN without adequate geologic storage capacity? Which regions emerge as most interesting due to variations in biomass availability? How does availability of low-cost wind resources drive DAC deployment in the Midwest, or other regions?

I attach two references which discuss variations of BECCS at regional scale across the United States. I did not see either cited by the authors. They are Baik et al. (2018) & Langholtz et al. (2020).

Suggested revisions:

-Biomass availability scenarios

It is common practice in bioenergy-focused capacity expansion modeling to consider variation in the biomass resource. Are the authors able to explore high/low estimates of biomass availability, and how that impacts BECCS deployment? Is biomass availability the primary driver of regional variation in BECCS deployment? What fraction of available biomass resource is utilization across different policy scenarios?

-Exploration of flexibility on decarbonization

This is an unexplored research need around BECCS and DAC in prior capacity expansion modeling. Potential questions include: Is BECCS solely a baseload resource, or is it able to be deployed in a more flexible manner? How does variation in this assumption affect its deployment? How is reliable power provided for DAC in regions where it is employed?

-Heat availability for DAC

What are REGEN's assumptions about availability of heat for DAC? Where is this heat provided? The paper makes brief mention of heat requirements for DAC but does not provide any further elaboration.

Reviewer #2 (Remarks to the Author):

The paper claims to offer the first detailed assessment of negative emission technologies on power systems operation and investment. The study poses an interesting question, deploys what appears to be an appropriate model for exploring it, and yields insightful results.

Throughout, I wondered how well the paper adhered to EPRI's own guidance about US-REGEN: "When viewing model results, it is important to keep in mind that analyses are not intended to be viewed as a prediction of a particular outcome or cluster of outcomes." Could the authors think about whether they should rebalance the focus between the broader lessons that can be learnt from this study and the specific numerical results from the scenario?

Line 90: Please cite the sources of data shown in this chart.

Figure 1: I would have thought the cost of DAC was a key consideration, could it be plotted alongside the other technologies (with \$/tCO₂ converted into \$/MW) to add context?

Line 96: Please could you provide some justification and context to your major study design choices? For example, why choose 80-140% reductions in CO₂? Why are 14 and 16 MMBTU/MWh your sensitivity values around a central number of 14.8? Please be clear if these were arbitrary choices or based on some underlying logic and reason.

Figure 3 and 4: Nature has very solid advice in their guidance for authors around making figures completely understandable standalone from the text. Please follow this, including more descriptive captions that explain what 95% cap means (easily mistaken for 95% capacity), and what an RPS is.

Line 194: It looks as though there is a fixed point at which DAC becomes more attractive than BECCS for marginal capacity additions, somewhere between 100% and 110%. Could you identify where that threshold sits and quantify it in terms of MtCO₂/year? I imagine it would be a useful value to know, as your findings suggest that BECCS is the preferred technology up to X, and DAC beyond it" – but with the current results, we don't quite know what X is.

I then wonder how sensitive is that threshold point to your input assumptions? If the costs of each technology vary within the range of the scenarios you consider, how much does that threshold change?

I would have expected a longer methods section in line with other Nature Communications papers. The current section barely scratches the surface of this study, and is insufficient to enable replication of the work. The reproducibility crisis within science is as present as ever, and your study could do more to combat this. <https://www.nature.com/news/1-500-scientists-lift-the-lid-on-reproducibility-1.19970>

Line 262, could you explain how US-REGEN works, as even the most basic details (is it linear, mixed integer, or some other form of optimization) are missing. What is the code availability?

Line 284: You explain where your renewable output time series data come from, but what about demand? Do you assume historic load profiles, in which case you miss some significant features of evolution which may occur (e.g. uptake of electric vehicles)? This is only hinted at later on in the appendix figures.

Line 284: Generating hourly time series of wind and solar PV output from meteorological data is non-trivial. You give no sources or explanation here, so this fails the test of scientific reproducibility. Could you please explain further about the methods employed and validation that was undertaken?

Line 284: How many weather years did you consider? Several papers from Europe have shown the importance of using multiple years of wind and solar data, you could consider these.

Line 480: Why is there a \$4 wheeling charge? Is this empirically defined, is it a theoretical assumption, is it a "fudge factor"?

Figure 9: How were the biomass supply curves derived? Given that you do not cite any sources in the figure caption, you implicitly suggest these results are original to your particular study. If that is the case, you need to provide some description of how these results were created. If this figure comes from previous work, you must mention the source in the caption otherwise you are bordering on plagiarism.

Figure 11: I am surprised you only have industrial sources for your data inputs. There have been many recent academic papers looking at the future cost of electrolysis and its performance, which could provide another angle to contextualize your data.

The supplementary results section is all very useful and well presented, no comments here.

Response to Reviewers

Reviewer #1

Comment 1.1: “This is a well-written, rigorous paper on an under-studied topic of interest to broad readership: the impacts of carbon removal technologies (BECCS and DAC) deployment on deep decarbonization of the U.S. electric power sector. The model, REGEN, is well-developed and well-documented. The range of policy scenarios tested are reasonable. The results are of value to numerous decision-makers. Nevertheless, I believe the representation of inputs, results, and analysis could be improved in many ways. Namely, there needs to be increased discussion of the geospatial aspects of deployment. Specifically, authors must consider and present analysis of variation of BECCS and DAC deployment and discuss factors with largest potential regional variation: geologic storage potential, regional biomass availability, and availability of low-cost renewable resources. These have been identified as potential drivers of variation in CDR deployment in prior studies (discussed below), but are noticeably lacking from the report, save for data reported in a single figure in the Appendix. I require this issue to be addressed prior to publication in Nature Communications. Based upon the information provided in Figure Figure 22, “Regional deployment of BECCS and DAC capacity by CDR availability scenario under the 140% CO₂ reduction scenario.” I believe this is a reasonable request. Possible questions to be addressed by the analysis & discussion could include: are there regions of REGEN without adequate geologic storage capacity? Which regions emerge as most interesting due to variations in biomass availability? How does availability of low-cost wind resources drive DAC deployment in the Midwest, or other regions? I attach two references which discuss variations of BECCS at regional scale across the United States. I did not see either cited by the authors. They are Baik et al. (2018) & Langholtz et al. (2020).”

Response 1.1: Thank you for your review and suggestions, all of which helped to improve the paper. We added new sensitivities to the analysis related to CO₂ storage and biomass supply to investigate these questions. A new section was added to the manuscript (Section 2.5 on “The Economic Geography of CDR”), which addresses the questions raised by the reviewer. Descriptions of these scenarios were added to the beginning of the “Results” section and “Methods.” We included references to the Baik, et al. (2018) and Langholtz, et al. (2020) papers in this section.

Comment 1.2: “Biomass availability scenarios: It is common practice in bioenergy-focused capacity expansion modeling to consider variation in the biomass resource. Are the authors able to explore high/low estimates of biomass availability, and how that impacts BECCS deployment? Is biomass availability the primary driver of regional variation in BECCS deployment? What fraction of available biomass resource is utilization across different policy scenarios?”

Response 1.2: We added biomass sensitivities, and the scenario assumptions are discussed in Section 2.1 (“Modeling Deep Decarbonization in the Electric Sector”) and Methods. The results are included in the new Section 2.5 (“The Economic Geography of CDR”).

Comment 1.3: “Exploration of flexibility: This is an unexplored research need around BECCS and DAC in prior capacity expansion modeling. Potential questions include: Is BECCS solely a baseload resource, or is it able to deployed in a more flexible manner? How does variation in this assumption affect its deployment? How is reliable power provided for DAC in regions where it is employed?”

Response 1.3: This is another good topic where our results offer insight. We added a new section (Section 2.6, “Operational Flexibility of CDR Technologies”) to address these questions with a couple new scenarios and new reporting for existing scenarios.

Comment 1.4: “Heat availability for DAC: What are REGEN's assumptions about availability of heat for DAC? Where is this heat provided? The paper makes brief mention of heat requirements for DAC but does not provide any further elaboration.”

Response 1.4: We added text to Section 2.1 and Methods to describe the heat requirements for DAC and the sources of assumptions used in the analysis.

Reviewer #2

Comment 2.1: “The paper claims to offer the first detailed assessment of negative emission technologies on power systems operation and investment. The study poses an interesting question, deploys what appears to be an appropriate model for exploring it, and yields insightful results. Throughout, I wondered how well the paper adhered to EPRI’s own guidance about US-REGEN: “When viewing model results, it is important to keep in mind that analyses are not intended to be viewed as a prediction of a particular outcome or cluster of outcomes.” Could the authors think about whether they should rebalance the focus between the broader lessons that can be learnt from this study and the specific numerical results?”

Response 2.1: Thank you for your comment and this observation. In this revision, we have sought to clarify and motivate key insights throughout the text while also connecting our conclusions to specific numerical results from the analysis.

Comment 2.2: “Line 90: Please cite the sources of data shown in this chart.”

Response 2.2: The caption for Figure 1 was updated to include data sources and additional text.

Comment 2.3: “Figure 1: I would have thought the cost of DAC was a key consideration, could it be plotted alongside the other technologies (with \$/tCO₂ converted into \$/MW) to add context?”

Response 2.3: We agree that the scenario assumptions for DAC and BECCS are important to document clearly given the focus of the study. We moved Table 1 into the body of the paper. Since the capacity costs for DAC are in \$ per t-CO₂ annual capture capacity terms, we added a comparison of BECCS costs in these same units to Table 1.

Comment 2.4: “Line 96: Please could you provide some justification and context to your major study design choices? For example, why choose 80-140% reductions in CO₂? Why are 14 and 16 MMBTU/MWh your sensitivity values around a central number of 14.8? Please be clear if these were arbitrary choices or based on some underlying logic and reason.”

Response 2.4: We added text to the body of the paper and Methods sections to justify the study’s scenario design choices and assumptions, specifically distinguishing between which sensitivities are based on studies and which are parametric given uncertainty about future values. In addition, we expanded the range of BECCS capital costs and heat rates considered for the sensitivities given the uncertainty about these values and range in the current literature.

Comment 2.5: “Figure 3 and 4: Nature has very solid advice in their guidance for authors around making figures completely understandable standalone from the text. Please follow this, including more descriptive captions that explain what 95% cap means (easily mistaken for 95% capacity), and what an RPS is.”

Response 2.5: We expanded the captions for all figures in the manuscript to make these more descriptive and help each figure to stand alone from the text.

Comment 2.6: “Line 194: It looks as though there is a fixed point at which DAC becomes more attractive than BECCS for marginal capacity additions, somewhere between 100% and 110%. Could you identify where that threshold sits and quantify it in terms of MtCO₂/year? I imagine it would be a useful value to know, as your findings suggest that BECCS is the preferred technology up to X, and DAC beyond it” – but with the current results, we don’t quite know what X is. I then wonder how sensitive is that threshold

point to your input assumptions? If the costs of each technology vary within the range of the scenarios you consider, how much does that threshold change?”

Response 2.6: We ran additional sensitivities in 5% increments to evaluate crossover points between BECCS and DAC under different technology assumptions. We added a figure in Appendix B to show these changes for BECCS and DAC deployment and added text in the second paragraph of Section 2.4 (“Sensitivity on CDR Demand for Non-Electric Residual Emissions”).

Comment 2.7: “I would have expected a longer methods section in line with other Nature Communications papers. The current section barely scratches the surface of this study, and is insufficient to enable replication of the work. The reproducibility crisis within science is as present as ever, and your study could do more to combat this.”

Response 2.7: We expanded the Methods section considerably to clarify the main assumptions across the scenarios and to provide greater transparency.

Comment 2.8: “Line 262, could you explain how US-REGEN works, as even the most basic details (is it linear, mixed integer, or some other form of optimization) are missing. What is the code availability?”

Response 2.8: We expanded the section discussing REGEN’s structure/assumptions and added a statement about data/code availability and separate “Data Availability”/“Code Availability” statements.

Comment 2.9: “Line 284: You explain where your renewable output time series data come from, but what about demand? Do you assume historic load profiles, in which case you miss some significant features of evolution which may occur (e.g. uptake of electric vehicles)? This is only hinted at later on in the appendix figures.”

Response 2.9: We added a paragraph to the Methods section to describe where the hourly load profiles come from and the main assumptions for model and scenario: “Hourly load profiles come from the REGEN end-use model...”.

Comment 2.10: “Line 284: Generating hourly time series of wind and solar PV output from meteorological data is non-trivial. You give no sources or explanation here, so this fails the test of scientific reproducibility. Could you please explain further about the methods employed and validation that was undertaken?”

Response 2.10: We added text to the Methods section to provide a high-level summary of the process used to generate and validate the hourly wind and solar output (“Hourly regional renewable output...”). Given the level of detail involved with this process (as the reviewer suggests), we also provide a link to the section in the full REGEN documentation that provides additional information about this modeling.

Comment 2.11: “Line 284: How many weather years did you consider? Several papers from Europe have shown the importance of using multiple years of wind and solar data, you could consider these.”

Response 2.11: This is an important area for further research. We clarify in the expanded Methods section that the regional hourly profiles for renewables and load (driven by temperature) are based on a single year (2015 in this case), but comment on the implication of considering multiple years. We also point out that the model includes a reserve constraint on local capacity adequacy excluding variable renewables, so the findings are reasonably robust to more extreme wind and solar droughts.

Comment 2.12: "Line 480: Why is there a \$4 wheeling charge? Is this empirically defined, is it a theoretical assumption, is it a "fudge factor"?"

Response 2.12: We have clarified that this cost is intended as an artificial proxy for uncaptured operational barriers to inter-regional power transfers.

Comment 2.13: "Figure 9: How were the biomass supply curves derived? Given that you do not cite any sources in the figure caption, you implicitly suggest these results are original to your particular study. If that is the case, you need to provide some description of how these results were created. If this figure comes from previous work, you must mention the source otherwise you are bordering on plagiarism."

Response 2.13: We added a paragraph to the Methods section to summarize how the biomass supply curves are derived. We also created a more descriptive caption for the biomass supply curve figure.

Comment 2.14: "Figure 11: I am surprised you only have industrial sources for your data inputs. There have been many recent academic papers looking at the future cost of electrolysis and its performance, which could provide another angle to contextualize your data."

Response 2.14: We conducted a more comprehensive literature review of electrolysis cost projections and updated this appendix figure accordingly.

Comment 2.15: "The supplementary results section is all very useful and well presented, no comments."

Response 2.15: Thank you. Note that we moved some of the material from SI Appendix A into the Methods section, per Response 2.7.

REVIEWERS' COMMENTS

Reviewer #1 (Remarks to the Author):

I thank the authors for adding information regarding the "economic geography" and operational flexibility of BECCS and DAC, along with comparison to two important previous papers on this topic (Baik et al / Langholtz et al). Inclusion of this information has greatly improved the paper. I recommend publication.

Reviewer #2 (Remarks to the Author):

My apologies for the delay in re-reviewing this manuscript. The authors had made extensive changes to the manuscript that warranted a thorough re-read.

The authors have considered each of my comments I adapted the manuscript in an appropriate way. I am left with no follow-up questions about these, as the authors have made thorough and diligent improvements.

I only have one remaining query. On page 14, have the authors left in some draft XX's in place of numbers? "Under typical assumptions about heat rates and emissions factors, a BECCS unit would receive approximately a \$X/MWh payment for carbon removal for a \$X/t-CO₂ carbon price."

Response to Reviewers

Reviewer #1

Comment 1.1: "I thank the authors for adding information regarding the "economic geography" and operational flexibility of BECCS and DAC, along with comparison to two important previous papers on this topic (Baik et al / Langholtz et al). Inclusion of this information has greatly improved the paper. I recommend publication."

Response 1.1: Thank you again for your helpful suggestions.

Reviewer #2

Comment 2.1: “My apologies for the delay in re-reviewing this manuscript. The authors had made extensive changes to the manuscript that warranted a thorough re-read. The authors have considered each of my comments I adapted the manuscript in an appropriate way. I am left with no follow-up questions about these, as the authors have made thorough and diligent improvements. I only have one remaining query. On page 14, have the authors left in some draft XX’s in place of numbers? ‘Under typical assumptions about heat rates and emissions factors, a BECCS unit would receive approximately a \$X/MWh payment for carbon removal for a \$X/t-CO₂ carbon price.’”

Response 2.1: Thank you again for your useful comments. We modified this sentence: “Under typical assumptions about heat rates and emissions factors, a BECCS unit would receive approximately a \$1/MWh payment for carbon removal for each \$1/t-CO₂ carbon price.”